# Mycolic acid-specific T cells protect against *Mycobacterium tuberculosis* infection in a humanized transgenic mouse model

Jie Zhao, Sarah Siddiqui, Shaobin Shang, Yao Bian, Sreya Bagchi, Ying He, Chyung-Ru Wang*

Department of Microbiology and Immunology, Northwestern University, Chicago, United States

**Abstract** Group 1 CD1 molecules, CD1a, CD1b and CD1c, present lipid antigens from *Mycobacterium tuberculosis* (Mtb) to T cells. Mtb lipid-specific group 1 CD1-restricted T cells have been detected in Mtb-infected individuals. However, their role in protective immunity against Mtb remains unclear due to the absence of group 1 CD1 expression in mice. To overcome the challenge, we generated mice that expressed human group 1 CD1 molecules (hCD1Tg) and a CD1b-restricted, mycolic-acid specific TCR (DN1Tg). Using DN1Tg/hCD1Tg mice, we found that activation of DN1 T cells was initiated in the mediastinal lymph nodes and showed faster kinetics compared to Mtb Ag85B-specific CD4[+] T cells after aerosol infection with Mtb. Additionally, activated DN1 T cells exhibited polyfunctional characteristics, accumulated in lung granulomas, and protected against Mtb infection. Therefore, our findings highlight the vaccination potential of targeting group 1 CD1-restricted lipid-specific T cells against Mtb infection.

*For correspondence: chyung-ru-wang@northwestern.edu

**Competing interests:** The authors declare that no competing interests exist.

## Introduction

The CD1 family of antigen presenting molecules presents self and microbial lipids to T cells (*Van Rhijn et al., 2013*; *De Libero and Mori, 2014*; *Adams, 2014*). Two major groups of CD1 iso-forms have been identified in humans: group 1 CD1 (CD1a, CD1b, and CD1c) and group 2 CD1 (CD1d) (*Adams, 2014*). While CD1d is broadly expressed, group 1 CD1 expression is limited to cortical thymocytes and professional antigen presenting cells (*Barral and Brenner, 2007*). Humans express all CD1 isoforms while mice only express CD1d. Due to the lack of a suitable small animal model to study group 1 CD1-restricted T cells, CD1d-restricted NKT cells have been better studied. A large proportion of CD1d-restricted NKT cells express an invariant TCR α chain and are known as iNKT cells (*Bendelac et al., 2007*). Unlike conventional T cells, which are positively selected by thymic epithelial cells (TECs), hematopoietic cells (HCs) select iNKT cells (*Bendelac et al., 2007*). The unique developmental selection program is thought to drive their pre-activated phenotype, which allows for rapid effector function manifestations upon TCR stimulation (*Bendelac et al., 2007*).

Unlike iNKT cells, group 1 CD1-restricted T cells are known to have diverse TCR usage (*Grant et al., 1999*; *Vincent et al., 2005*; *Felio et al., 2009*). Group 1 CD1-restricted T cell responses have mostly been characterized in the context of mycobacterial antigens, although recent studies have shown that humans have a significant proportion of autoreactive group 1 CD1-restricted T cells (*de Jong et al., 2010*; *de Lalla et al., 2011*). The Mtb cell wall is lipid rich (60% of its cell wall is composed of lipids) and contains a plethora of lipid antigens that are presented by group 1 CD1 molecules (*Van Rhijn et al., 2013*; *De Libero and Mori, 2014*). Among group 1 CD1

**eLife digest** Most cases of tuberculosis are caused by a bacterium called *Mycobacterium tuberculosis*, which is believed to have infected one third of the world's population. Most of these infections are dormant and don't cause any symptoms. However, active infections can be deadly if left untreated and often require six months of treatment with multiple antibiotics. One reason why these infections are so difficult to treat is because the *M. tuberculosis* cell walls contain fatty molecules known as mycolic acids, which make the bacteria less susceptible to antibiotics. These molecules also help the bacteria to subvert and then hide from the immune system.

The prevalence of the disease and the increasing problem of antibiotic resistance have spurred the search for an effective vaccine against tuberculosis. While most efforts have focused on using protein fragments in tuberculosis vaccines, some evidence suggests that human immune cells can recognize fatty molecules such as mycolic acids and that these cells could help manage and control *M. tuberculosis* infections. However, it has been difficult to determine whether these immune cells genuinely play a protective role against the disease because most vaccine research uses mouse models and mice do not have an equivalent of these immune cells.

Now, Zhao et al. have engineered a "humanized" mouse model that produces the fatty molecule-specific immune cells, and show that these mice do respond to the presence of mycolic acids. Infecting the genetically engineered mice with *M. tuberculosis* revealed that the fatty molecule-specific immune cells were quickly activated within lymph nodes at the center of the chest. These cells later accumulated at sites in the lung where the bacteria reside, and ultimately protected against *M. tuberculosis* infection. The results show that these specific immune cells can counteract *M. tuberculosis*, and highlight the potential of using mycolic acids to generate an effective vaccine that provides protection against tuberculosis.

molecules, CD1b presents the largest pool of Mtb-derived lipids like mycolic acid (MA), glucose monomycolate, glycerol monomycolate, diacylated sulfoglycolipids, lipoarabinomannan and phosphatidylinositol mannoside to cognate T cells (*Van Rhijn et al., 2013*; *De Libero and Mori, 2014*). Of the Mtb lipids mentioned above, MAs are the major lipid constituents of the Mtb cell envelope and considered a potent Mtb virulence factor (*Barry et al., 1998*; *Karakousis et al., 2004*). Interestingly, MA-specific CD1b-restricted T cells have been detected in the blood as well as disease sites of Mtb-infected individuals (*Montamat-Sicotte et al., 2011*).

Tuberculosis (TB), the disease caused by Mtb, is a global health burden, especially in developing countries and amongst HIV/AIDS patients. Additionally, due to the emergence of multidrug-resistant Mtb and the lack of an effective vaccine to prevent pulmonary TB in adults, it is important to decipher the role of various T cell subsets in Mtb infection for the development of better preventive or therapeutic vaccines (*Ottenhoff et al., 2012*). The subunit vaccines currently under development for Mtb utilize peptide or protein antigens which target MHC-restricted conventional T cells (*Dorhoi and Kaufmann, 2014*), but the utility in targeting lipid antigens has not been explored. Since CD1 molecules are non-polymorphic, CD1-restricted Mtb lipid antigens are likely to be recognized by most individuals, making them attractive vaccine targets (*Barral and Brenner, 2007*). Several lines of evidence suggest that Mtb lipid-specific group 1 CD1-restricted T cells contribute to anti-mycobacterial immunity. Investigation of group 1 CD1-restricted T cell lines derived from healthy or mycobacteria-infected individuals has revealed that these T cells are cytotoxic and produce IFN-γ and TNF-α, cytokines critical for protective immunity to TB (*Van Rhijn et al., 2013*). Moreover, group 1 CD1-restricted Mtb lipid-specific T cells are found in higher frequencies in individuals exposed to Mtb compared with control populations, suggesting that they are activated following infection with Mtb (*Moody, 2000*; *Ulrichs et al., 2003*; *Gilleron, 2004*; *Layre et al., 2009*; *Moody et al., 2000*). Additionally, a robust Mtb lipid-specific group 1 CD1-restricted T cell response has been detected in Mtb-infected human group 1 CD1 transgenic mice (*Felio et al., 2009*). However, it remains unclear whether this unique T cell subset plays a protective role during the course of infection.

In this study, we generated transgenic mice expressing mycolic acid-specific CD1b-restricted TCR (DN1Tg) and human group 1 CD1 molecules (hCD1Tg). Using this mouse model, we found that

DN1 T cells were selected most efficiently by group 1 CD1-expressing HCs in the thymus. Upon adoptive transfer of DN1 T cells to Mtb-infected hCD1Tg mice, DN1 T cells were first activated in the mediastinal lymph nodes, exhibiting faster kinetics than Ag85B-specific CD4$^+$ T cells. DN1 T cells were cytotoxic, polyfunctional and contributed to anti-mycobacterial immunity by reducing bacterial burdens in the lung, spleen and liver. Thus, this study provides the first direct demonstration that group 1 CD1-restricted Mtb lipid-specific T cells play a protective role during Mtb infection.

## Results

### Generation of a mycolic acid-specific CD1b-restricted TCR transgenic mouse model

We developed human CD1 transgenic mice, which expressed group 1 CD1 molecules in a similar pattern to that observed in humans. Using this model, we demonstrated the feasibility to study group 1 CD1-restricted T cell responses in aerosol infection with Mtb (*Felio et al., 2009*). To facilitate the direct analysis of Mtb lipid-specific group 1 CD1-restricted T cells, we generated a novel transgenic mouse strain that expressed a human/mouse chimeric TCR, composed of variable region from human T cell clone DN1 (*Grant et al., 1999*), specific for CD1b/mycolic acid (MA), and mouse TCR constant region (*Figure 1A*). DN1Tg founders and their progeny were screened for the presence of *TRAV13-2-TRAJ57* gene fragment by PCR and for the surface expression of human Vβ5.1 (TRBV5-1) by flow cytometry (*Figure 1B,C*). Subsequently, DN1Tg mice were bred onto hCD1Tg/ Rag$^{-/-}$ background to eliminate the expression of endogenous TCR. All DN1Tg mice used in this study were on a Rag$^{-/-}$ background. To examine whether the development of DN1 T cells was dependent on group 1 CD1 molecules, we compared DN1 T cells in WT and hCD1Tg backgrounds. We found that both frequency and absolute number of DN1 T cells were greatly reduced in DN1Tg mice compared with DN1Tg/hCD1Tg mice in all tested organs (*Figure 1D–F*). This suggested that group 1 CD1 supported the development of DN1 T cells. Notably, unlike CD1d-restricted iNKT cells, DN1 T cells from the spleen and lymph nodes of DN1Tg/hCD1Tg mice exhibited a naïve phenotype (characterized by low expression levels of T cell activation markers such as CD69 and CD44) similar to conventional CD8$^+$ T cells and were either CD8αβ$^+$ or CD4$^-$CD8$^-$ (DN). In addition, DN1 thymocytes from DN1Tg/hCD1Tg mice did not express PLZF, the master transcription factor for innate T cell lineages (*Figure 1G*) (*Kovalovsky et al., 2008*; *Savage et al., 2008*).

### CD1b-expressing hematopoietic cells (HCs) most efficiently select DN1 T cells

Unlike conventional T cells, which are positively selected by TECs, iNKT cells are exclusively selected by CD1d-expressing thymocytes (*Bendelac, 1995*; *Coles and Raulet, 2000*). Several studies have demonstrated the correlation between positive selection on HCs and a pre-activated T cell phenotype of innate-like T cells (*Bendelac et al., 2007*; *Cho et al., 2011*; *Bediako et al., 2012*). Given that DN1 T cells exhibited a naïve surface phenotype, one would expect DN1 T cells to be positively selected by TECs. To test this hypothesis, we adoptively transferred bone marrow from DN1Tg and DN1Tg/hCD1Tg mice (in the Rag-deficient background) into irradiated CD45.1 congenic WT and hCD1Tg recipients. 5 weeks after transfer, DN1 T cells were identified by CD45.2and hVβ5.1 surface expression in different groups (*Figure 2A*). The percentage (*Figure 2B*) and absolute number (*Figure 2C*) of DN1 T cells were significantly higher in mice with group 1 CD1-expressing HCs compared to mice that only had group 1 CD1-expressing TECs. This suggested that HCs most efficiently mediate the positive selection of DN1 T cells. As a small number of DN1 T cells developed in mice that lack CD1b (*Figure 2A*), it is possible that mouse CD1d is responsible for their selection. We compared the percentage of DN1 T cells in the spleen and thymus of DN1Tg/hCD1Tg (CD1d$^+$), DN1Tg/hCD1Tg/CD1d$^{-/-}$, DN1Tg (CD1d$^+$), and DN1Tg/CD1d$^{-/-}$ mice (all in the Rag-deficient background). We found that the percentage of DN1 T cells was comparable in DN1Tg/hCD1Tg and DN1Tg/hCD1Tg/CD1d$^{-/-}$ mice. In addition, DN1 T cells were barely detectable in the thymus and spleen of DN1Tg and DN1Tg/CD1d$^{-/-}$ mice. These data suggest that CD1d does not contribute to the thymic selection of DN1 T cells (*Figure 2—figure supplement 1*).

Comparing CD1b expression on TECs and CD4$^+$CD8$^+$ (DP) thymocytes revealed that DP thymocytes express significantly higher levels of CD1b than TECs (*Figure 2D*). Thus, CD1b-expressing

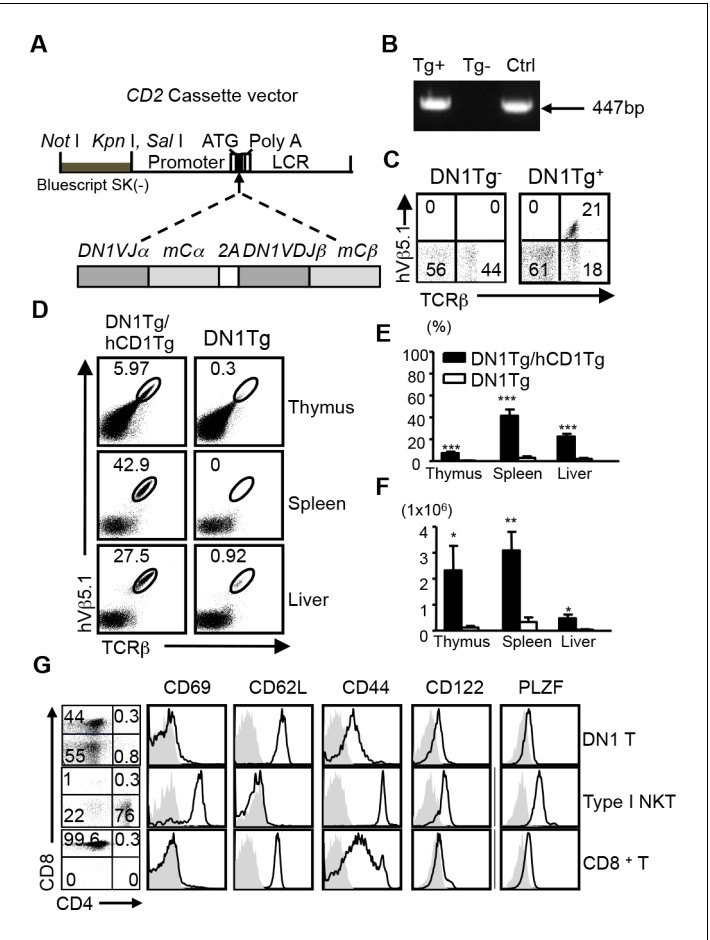

**Figure 1.** Development of DN1 T cells is dependent on the presence of group 1 CD1 molecules. (**A**) Schematic diagram of *DN1 TCR* construct used to generate DN1Tg mice. (**B**) The presence of *DN1 TCR* in the genomic DNA of transgenic mice was examined by PCR using primers specific for *TRAV13-2* and *TRAJ57*. *DN1* plasmid was used as a positive control (Ctrl). (**C**) DN1 T cells in the spleen of DN1Tg$^+$ and DN1Tg$^-$ mice (in a B6 background) were detected by FACS using anti-mouse TCRβ and anti-human Vβ5.1 mAbs. (**D**) Lymphocytes from the thymus, spleen and liver of DN1Tg/hCD1Tg and DN1Tg mice (in the Rag-deficient background) were analyzed for the presence of DN1 T cells (TCRβ$^+$hVβ5.1$^+$). (**E, F**) Bar graphs depict the mean and SEM of the percentages (in the lymphocyte gate) and absolute numbers of DN1 T cells from DN1Tg/hCD1Tg and DN1Tg mice (n=3–8 per group). ***p<0.001; **p<0.01; *p<0.05. (**G**) Expression of indicated markers (black line) on DN1 T cells (TCRβ$^+$hVβ5.1$^+$) from DN1Tg/hCD1Tg/Rag$^{-/-}$ mice, type I NKT cells (CD1d/αGalCer tetramer$^+$TCRβ$^+$) from WT mice, and conventional CD8$^+$ T cells (TCRβ$^+$CD8$^+$) from WT mice, compared with isotype control (gray filled). The expression of CD4 and CD8 on DN1 T cells and type I NKT cells were shown in the dot plots. Cells isolated from the thymus were used for PLZF staining. Results are representative of 3 experiments.

thymocytes may be better suited to mediate the positive selection of DN1 T cells. Since DN1 T cells could also be selected by TECs, albeit with much lower efficiency compared to HCs, we compared the phenotype of DN1 T cells that developed in mice expressing CD1b on both HC and TEC, HC only and TEC only. DN1 T cells in the periphery of these three groups had a comparable proportion of CD8$^+$/DN T cells (*Figure 2E*). In addition, DN1 T cells in the thymus of these three groups expressed similar levels of PLZF and CD44 (*Figure 2F*). However, DN1 T cells selected by HCs expressed higher levels of CD5 (*Figure 2F*), a surrogate marker for the TCR signaling strength in developing thymocytes, suggesting they might receive stronger TCR signals.

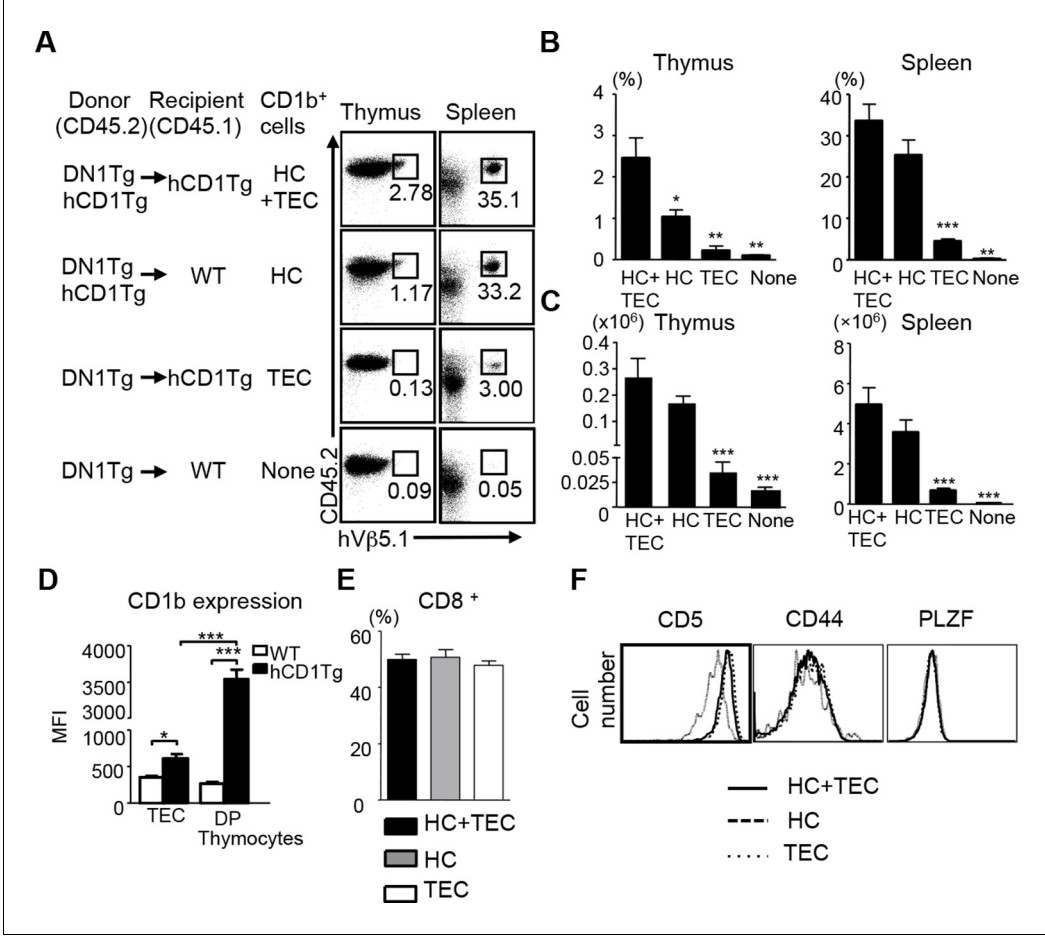

**Figure 2.** CD1b-expressing hematopoietic cells are the major cell type that medicates the positive selection of DN1 T cells. Bone marrow from DN1Tg and DN1Tg/hCD1Tg mice (in the Rag-deficient background) were adoptively transferred into irradiated CD45.1 congenic WT and hCD1Tg recipients and the development of DN1 T cells were examined 5 weeks later. (A) Dot plots depict the proportion of CD45.2[+] hVβ5.1[+] cells in the lymphocyte gate. Data are representative of 2 experiments with 4–6 mice in each group. (B, C) Bar graphs depict the mean ± SEM of the percentage and absolute number of CD45.2[+] hVβ5.1[+] cells in each group. Statistical significance was evaluated by comparing HC, TEC and None group with HC+TEC group. ***p<0.001; **p<0.01; *p<0.05. (D) CD1b expression on TEC (CD45[-]MHCII[+]) and DP thymocytes from WT and hCD1Tg mice were shown as MFI values (n=3 per group). (E) Percentage of CD8[+] DN1 T cells in the spleen of HC+TEC, HC or TEC groups of mice. (F) Expression of indicated markers on DN1 thymocytes that developed in HC+TEC, HC or TEC groups. ***p<0.001; **p<0.01; *p<0.05. Results are representative of 2 experiments with 3 mice per group.

The following figure supplement is available for figure 2:

**Figure supplement 1.** CD1 expression does not significantly affect the development of DN1 T cells.

## DN1 T cells exhibit effector functions in response to mycolic acid stimulation and Mtb infection

The human DN1 T cell line had been shown to secrete Th1 cytokines when stimulated with MA presented by CD1b[+] APCs (*Beckman et al., 1994*). To test whether DN1 T cells that developed in DN1Tg/hCD1Tg mice retained the same functional properties as the original human T cell line, we stimulated lymph node cells from DN1Tg/hCD1Tg mice with un-pulsed or MA-pulsed bone marrow derived dendritic cells (BMDCs) to detect IFN-γ production and antigen-specific cytotoxicity (*Figure 3A,B*). DN1 T cells produced IFN-γ in response to MA-pulsed hCD1Tg DCs but not WT DCs or un-pulsed DC, suggesting the activation of DN1 T cells required both antigen and group 1 CD1 molecules. In addition, DN1 T cells showed cytotoxic activity against MA-pulsed hCD1Tg DCs. MA is

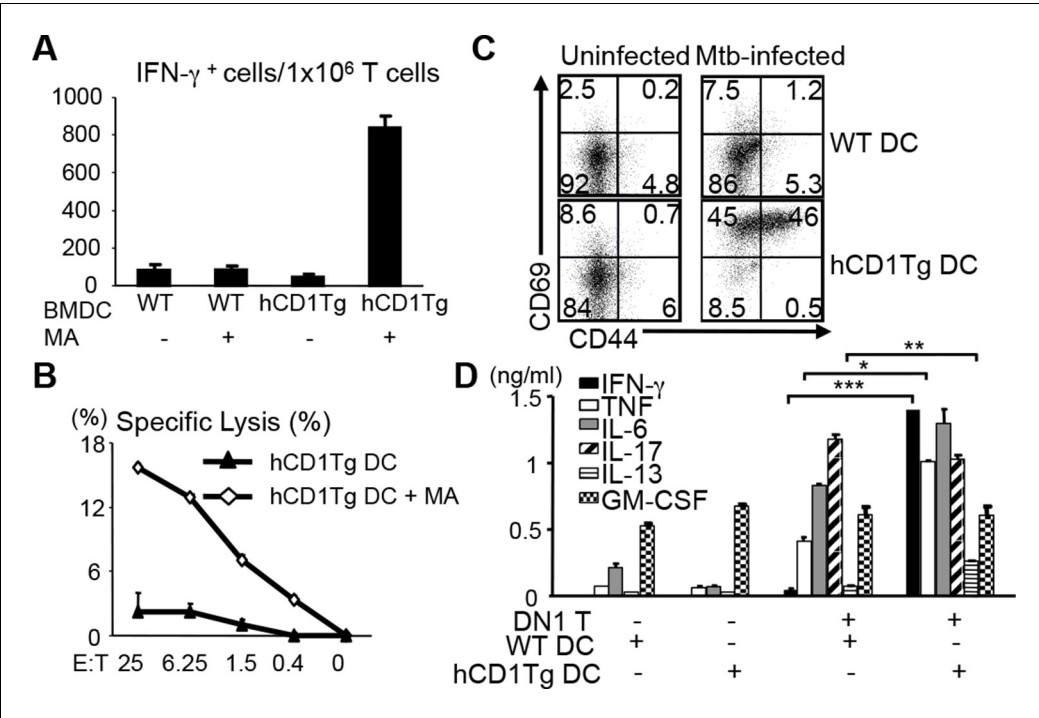

**Figure 3.** DN1 T cells acquire effector functions in response to MA-pulsed DC and Mtb-infected DC. (**A**) DN1 T cells isolated from lymph nodes of DN1Tg/hCD1Tg/Rag$^{-/-}$ mice were co-cultured with un-pulsed or MA-pulsed WT or hCD1Tg DC and IFN-γ producing cells were determined by ELISPOT assays. (**B**) DN1 T cells were stimulated by hCD1Tg BMDCs pulsed with MA for 7 days and then tested for cytotoxic activity against hCD1Tg BMDCs with or without MA. Data are representative of 3 experiments (mean ± SEM of triplicate cultures). (**C, D**) DN1 T cells isolated from lymph nodes of DN1Tg/hCD1Tg/Rag$^{-/-}$ mice were co-cultured with Mtb-infected BMDC for 48 hr. Activation markers on DN1 T cells were detected by flow cytometry and cytokines in the supernatant were detected by CBA flex set. ***p<0.001; **p<0.01; *p<0.05. Results are representative of 2 experiments with 3 mice per experiment.

located within the Mtb cell wall, either covalently attached via arabinogalactan to the cell wall peptidoglycan, or non-covalently associated in the form of trehalose dimycolate (*Barry et al., 1998*; *Karakousis et al., 2004*). To determine whether DN1 T cells can be activated by naturally processed MA, we set up co-culture of DN1 T cells with Mtb-infected BMDCs. As shown in *Figure 3C*, co-culture of DN1 T cells with Mtb-infected hCD1Tg DCs led to up-regulation of activation markers CD69 and CD44 on DN1 T cells. Furthermore, DN1 T cells produced multiple cytokines, of which the production of IFN-γ, TNF-α, IL-13 and IL-6 was dependent on the presence of group 1 CD1 molecules (*Figure 3D*), with the exception of IL-17. It is possible that co-stimulatory molecules and/or cytokines induced by Mtb-infected DCs can stimulate DN1 T cells to secrete this cytokine independent of TCR-CD1b interaction. Collectively, our data demonstrated that DN1 T cells became activated and exhibited effector functions in response to MA-pulsed or Mtb-infected DC in a group 1 CD1-dependent manner.

## DN1 T cell-mediated control of Mtb is dependent on antigen-presentation by group 1 CD1-expressing DCs and IFN-γ production

Macrophages are known as primary host cells for Mtb. Whereas BMDCs and a subset of myeloid DCs from hCD1Tg mice expressed high levels of CD1b, the expression of CD1b was almost undetectable on bone marrow derived macrophages (BMDMs) (*Figure 4A*), similar to the observation in human monocyte derived macrophages. Accordingly, we detected only minimal DN1 T cell activation when stimulated with Mtb-infected BMDMs (*Figure 4B*). While our data showed that Mtb-infected macrophages do not directly present antigen to DN1 T cells, apoptotic vesicles released from Mtb-infected macrophages have been shown to transfer mycobacterial antigens to uninfected

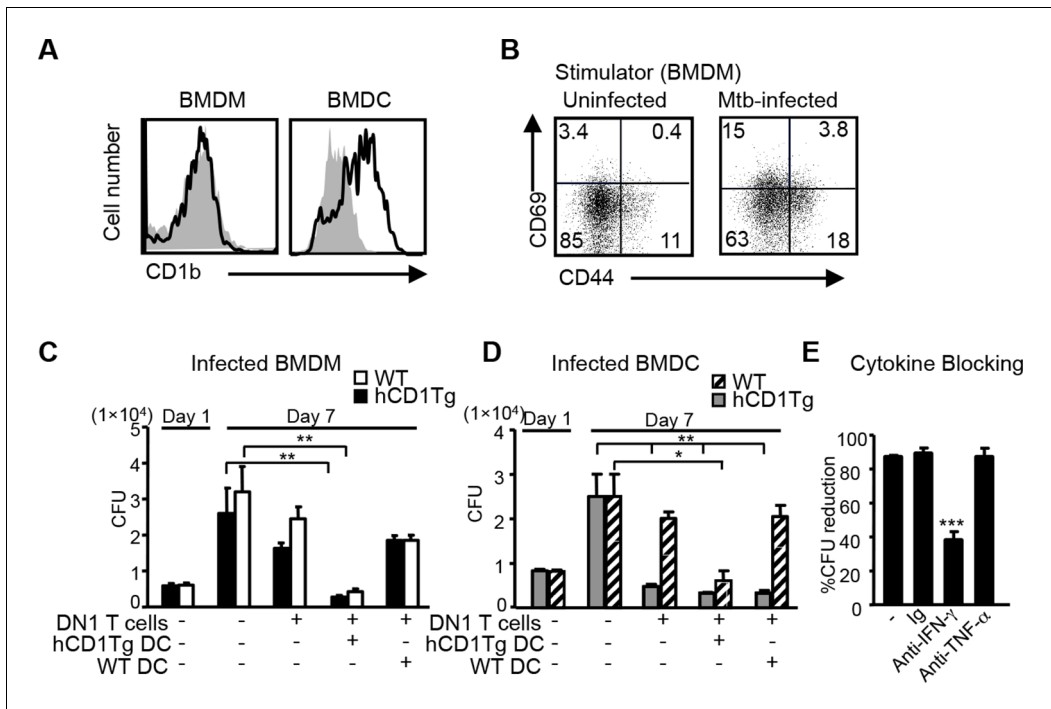

**Figure 4.** DN1 T cell-mediated control of Mtb is dependent on the antigen presentation by group 1 CD1-expressing DCs and IFN-γ production. (A) CD1b expression on BMDM and BMDC was detected using flow cytometry. (B) BMDMs were in vitro infected with Mtb (MOI=5) and DN1 T cells were added 1 day after infection. After 48 hr co-culture, activation markers on DN1 T cells were detected by flow cytometry. (C, D) WT and hCD1Tg BMDMs and BMDCs were infected with Mtb. 1 day later, DN1 T cells with or without uninfected WT or hCD1Tg DCs were added into the culture for another 6 days. At day 7 post-infection, cells were lysed for CFU assay. Bar represents mean and SEM from replicate cultures (n = 6). (E) DN1 T cells were added into Mtb-infected BMDCs in the presence of control Ig (Ig), anti-IFN-γ or anti-TNFα. At day 7 post-infection, cells were lysed for CFU assay. % reduction was calculated as 100x[(BMDC_alone - BMDC_with DN1)/ BMDC_alone]. Results are representative of 2–3 experiments. ***p<0.001; **p<0.01; *p<0.05.

APCs, such as DCs (*Ulrichs et al., 2003*; *Schaible et al., 2003*), which could in turn activate DN1 T cells. To explore whether and how DN1 T cells can control Mtb in infected macrophages, Mtb-infected BMDMs from WT or hCD1Tg mice were cultured together with DN1 T cells in the presence or absence of uninfected WT or hCD1Tg DCs. After 7 days, we determined the number of colony forming units (CFU) to investigate whether DN1 T cells inhibited intracellular bacterial growth within BMDMs. As expected, addition of DN1 T cells alone did not have a significant effect on bacterial burdens in Mtb-infected BMDMs. However, when DN1 T cells were added together with uninfected hCD1Tg DCs to infected BMDMs, the number of CFU decreased significantly compared with controls (*Figure 4C*). For comparison, we also used Mtb-infected BMDCs from WT or hCD1Tg mice as targets. We found that DN1 T cells efficiently controlled Mtb growth within infected hCD1Tg DCs but not WT DCs. Similarly, if DN1 T cells and uninfected hCD1Tg DCs were added to Mtb-infected WT DCs, Mtb growth was significantly inhibited (*Figure 4D*). These data indicated that group 1 CD1-expressing DCs mediated activation of DN1 T cells, which in turn controlled bacterial growth not only in the group 1 CD1-expressing DCs but also in macrophage and group 1 CD1-negative DCs. In addition, through a cytokine-blocking assay, we found that IFN-γ but not TNF-α was crucial for mediating anti-mycobacterial function of DN1 T cells (*Figure 4E*).

## Activation of DN1 T cells is initiated in mediastinal lymph nodes (MLN) in response to aerosol Mtb infection

To study when and where group 1 CD1-restricted Mtb lipid-specific T cells are first presented with antigens after aerosol infection with Mtb, we adoptively transferred naïve DN1 T cells into CD45.1

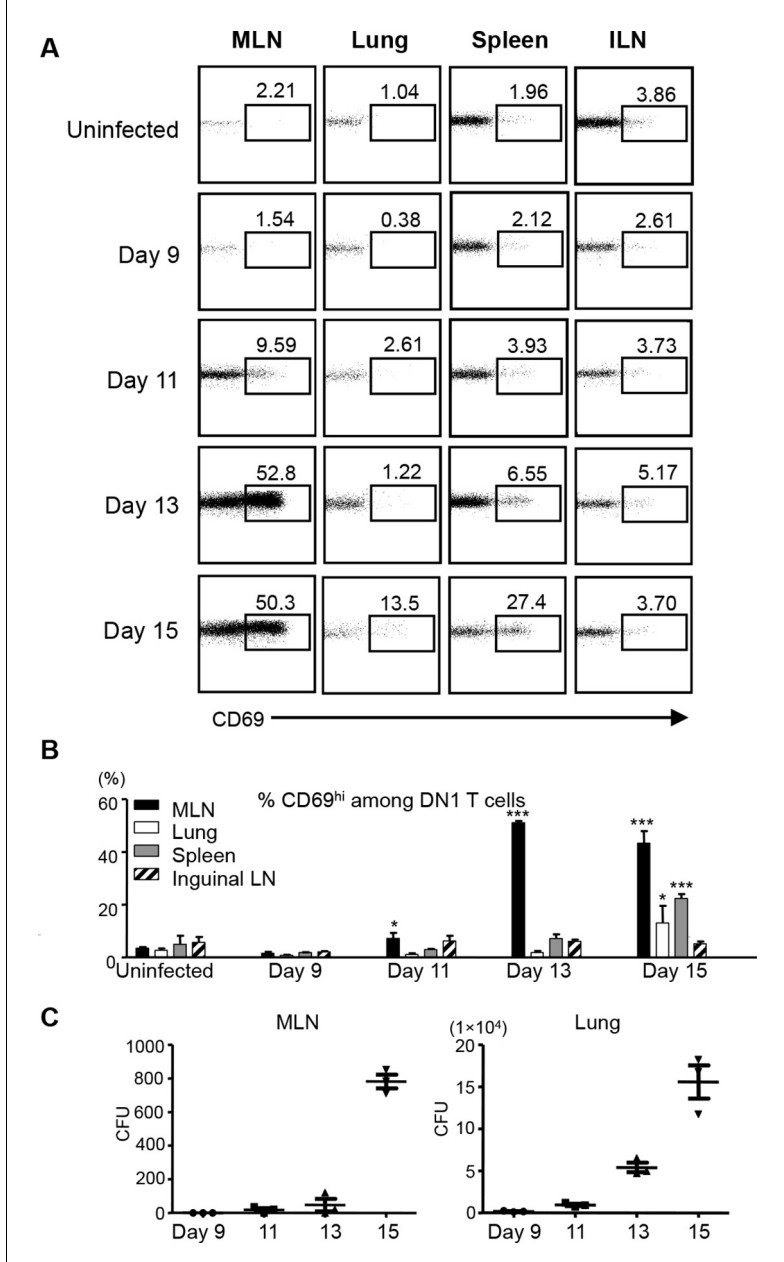

**Figure 5.** Activation of DN1 T cells is initiated in mediastinal lymph nodes after aerosol Mtb infection. (A) $3 \times 10^6$ naïve DN1 T cells were adoptively transferred into Mtb infected CD45.1 congenic hCD1Tg mice at day 7 post-infection. CD69 expression was detected on DN1 T cells (hVβ5$^+$TCRβ$^+$) from the MLN, lung, spleen, and inguinal lymph nodes (ILN) of recipient mice at indicated time points. (B) Bar graphs depict the mean and SEM of the percentages of CD69$^{hi}$ population among DN1 T cells (n=3 each time point). (C) Bacterial CFU in MLN and lung at indicated time points. Each symbol represents the bacteria burden in the MLN or lung of an individual mouse at the indicated time point. Horizontal bars represent the mean CFU counts ± SEM for each group. Results are representative of 2 experiments with 3 mice per time point. ***p<0.001; **p<0.01; *p<0.05.

congenic hCD1Tg mice that had been infected with Mtb 7 days earlier. The expression of CD69 on DN1 T cells from various organs was monitored. The up-regulation of CD69 on DN1 T cells was first observed as early as day 11 post-infection in the lung-draining MLN, but not in other tissues examined. By day 15 after infection, the activation of DN1 T cells was also detected in the lung and spleen (*Figure 5A,B*). Although bacterial burdens were much higher in the lung than in the MLN

(*Figure 5C*), DN1 T cell activation correlated with the first appearance of bacteria in MLN. Collectively, these data indicate that activation of DN1 T cells is initiated in MLN when Mtb disseminate from the site of primary infection (lung) to MLN.

## DN1 T cells are activated earlier compared with Ag85B-specific CD4$^+$ T cells after Mtb infection

Several studies have shown that activation of Mtb-specific CD4$^+$ conventional T cells is also initiated in the MLN (*Wolf et al., 2008*; *Reiley et al., 2008*; *Gallegos et al., 2008*). To compare the kinetics of DN1 T cell priming with conventional T cells after aerosol Mtb infection, we co-transferred naïve DN1 T cells with CD4$^+$ TCR transgenic T cells specific to a peptide from Mtb Ag85B (P25 T cells) to Mtb-infected mice to monitor their activation and proliferation. CellTrace Violet dye labeled P25 T cells and CFSE labeled DN1 T cells were co-transferred in equal numbers to CD45.1 congenic hCD1Tg mice that were infected 7 days earlier. Similar to the observation in *Figure 5A*, up-regulation of CD69 on DN1 T cells started at day 11 post infection while CD69 was up-regulated on a small percentage of P25 T cells in MLN 13 days after infection (*Figure 6A,C*). Additionally, cell division was detected on DN1 T cells by day 13 and on P25 T cells by day 15 in the MLN (*Figure 6B, D*). Also, compared to P25 T cells, a greater proportion of DN1 T cells in the lung and spleen expressed CD69 and underwent cell division at day 15 post-infection (*Figure 6E,F*). Taken together, our data suggests that activation of MA-specific CD1b-restricted T cells occurs earlier than Ag85B-specific MHC II-restricted CD4$^+$ T cells during Mtb infection.

## Adoptive transfer of DN1 T cells confers protection against Mtb

Group 1 CD1-restricted human T cell clones show effector functions including cytotoxic activity and cytokine production in response to Mtb-specific lipid antigens. However, whether group 1 CD1-restricted T cells confer protection against Mtb infection remains unknown. To address this question, DN1 effector T cells were adoptively transferred to hCD1Tg/Rag$^{-/-}$ mice and recipient mice were subsequently challenged with virulent Mtb via aerosol route. 4 weeks after infection, the number of bacteria in the lung, spleen and liver was determined. As shown in *Figure 7A*, DN1 T cells decreased the number of viable bacteria in hCD1Tg/Rag$^{-/-}$ recipient mice in all tested organs as compared to mice that received no DN1 T cell transfer. Moreover, adoptive transfer of DN1 T cells to Rag$^{-/-}$ recipient mice did not significantly reduce bacterial burdens suggesting that DN1 T cells confer protection in an hCD1Tg-dependent manner (*Figure 7A*). We also compared the protective capacity of DN1 T cells with non-relevant Listeria LemA-specific H2-M3-restricted D7 T cells (*Figure 7—figure supplement 1*). D7 T cells did not significantly reduce bacterial burdens in the lung of hCD1Tg/Rag$^{-/-}$ recipient mice compared to mice that received no T cell transfer (*Figure 7—figure supplement 1*).

DN1 T cells isolated from the lung of infected hCD1Tg/Rag$^{-/-}$ mice produced multiple cytokines (e.g. TNF-α, IFN-γ, and IL-2, *Figure 7B*) and expressed CD107a, a surrogate marker of cytotoxic activity (*Cho et al., 2011*), after ex vivo MA stimulation (*Figure 7C*). To further visualize the location and distribution of DN1 T cells, DN1 T cells in the lung were stained by immunohistochemistry using anti-CD3 antibody. A significantly higher number of DN1 T cells were seen within pulmonary granulomas of hCD1Tg/Rag$^{-/-}$ mice that received DN1 T cells compared to control groups (*Figure 7D,E*). In summary, these data demonstrate that DN1 T cells accumulate in granulomas and contribute to protective immunity against Mtb by producing multiple Th1-related cytokines and exerting cytotoxicity.

## Discussion

In this study we used a novel transgenic mouse model to examine the role of MA-specific CD1b-restricted T cells during Mtb infection. This model not only allowed for the in vivo tracking of Mtb lipid antigen-specific T cells but also for deciphering their developmental requirements and functional role in Mtb infection. Like CD1d-restricted NKT cells, DN1 T cells were most efficiently selected by HCs even though they did not exhibit a pre-activated phenotype. In both in vitro and in vivo Mtb infection systems, the presence of DN1 T cells resulted in lower bacterial burden, indicating their role in protective immunity against Mtb. Interestingly, DN1 T cells were first activated in the lung draining lymph node and seemed to have faster activation and proliferation kinetics

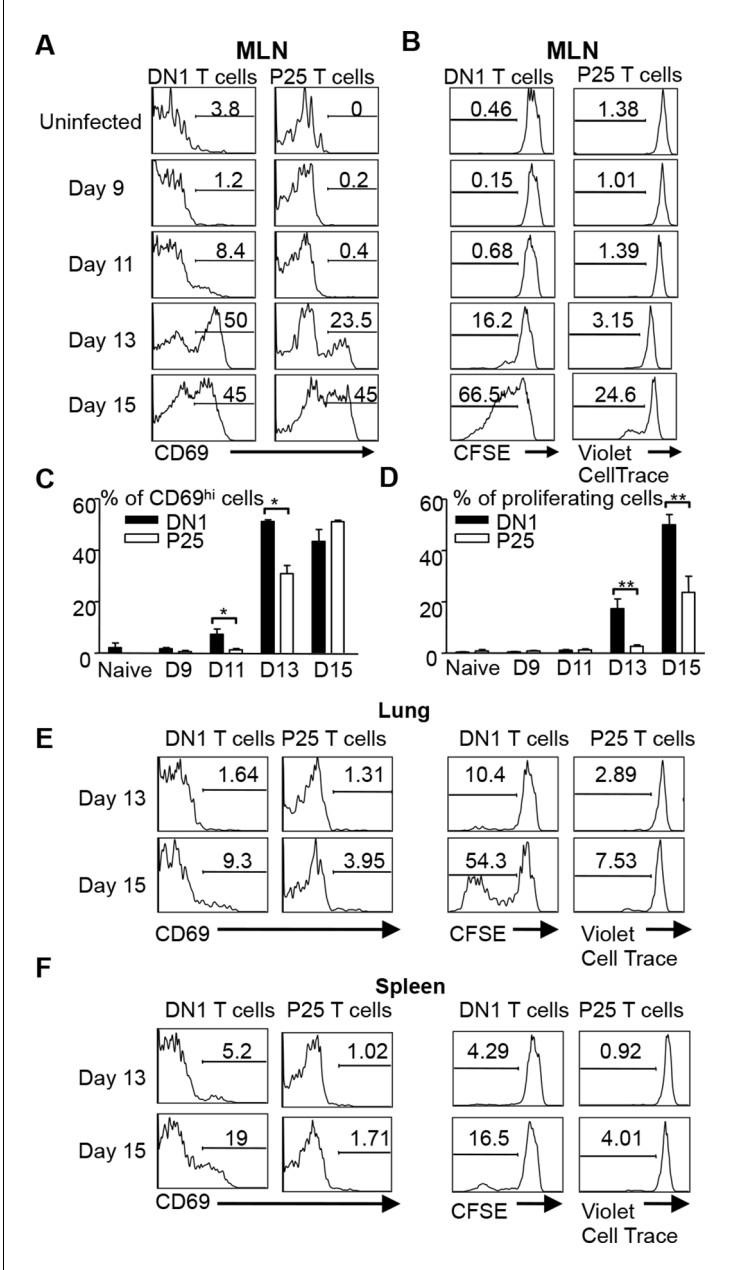

**Figure 6.** DN1 T cells are activated earlier than Ag85 specific CD4[+] T cells after Mtb infection. (**A, B**) 3x10[6] CFSE-labeled DN1 T cells and 3x10[6] CellTrace Violet-labeled P25 T cells were co-transferred into Mtb infected CD45.1 congenic hCD1Tg mice. CD69 expression, CFSE and CellTrace Violet were detected on DN1 T cells and P25 T cells from MLN at indicated time points. (**C, D**) Bar graphs depict the mean and SEM of the percentages of CD69[hi] and CFSE/Violet[low] populations among DN1 and P25 T cells. (**E, F**) CD69 expression, CFSE and CellTrace Violet were detected on DN1 and P25 T cells from lungs (**E**) and spleens (**F**) at day 13 and day 15 post-infection. Results are representative of 2 experiments with 3–4 mice per experiments. ***p<0.001; **p<0.01; *p<0.05.

compared to Ag85B-specific CD4[+] T cells. The distinct kinetics of the priming of these two T cell populations may allow them to contribute to anti-mycobacterial activity at the different stages of infection. However, since only one TCR was investigated in this study, it remains unclear whether these findings could be extrapolated to all group 1 CD1-restricted mycolic acid specific T cells.

While TECs select conventional T cells, innate-like T cells, characterized by an effector/memory phenotype in the naïve state, have been correlated with their selection by HCs (***Bendelac et al.,***

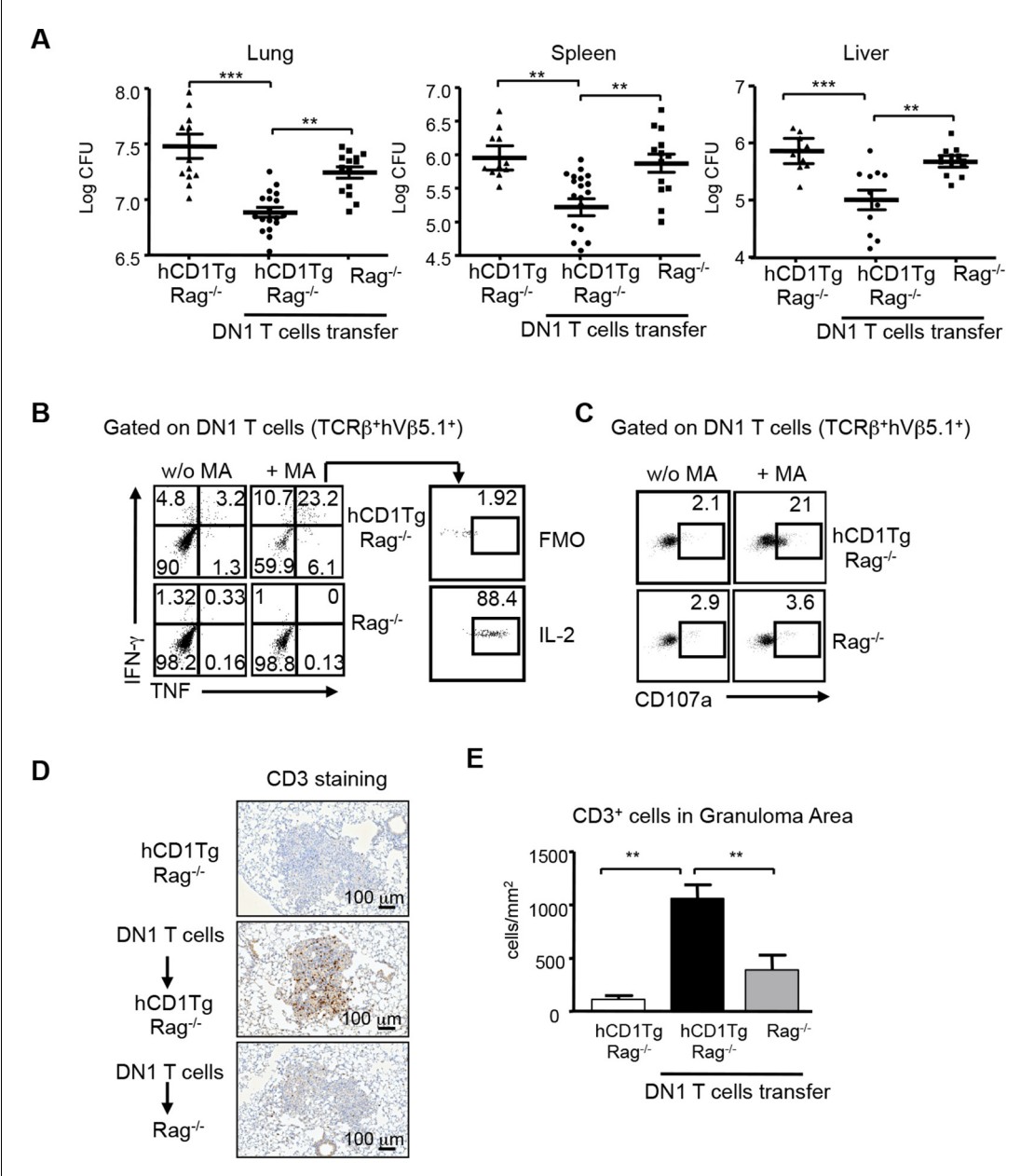

**Figure 7.** DN1 T cells contribute to protective immunity against Mtb infection. Effector DN1 T cells (5-7x10^6 cells) were transferred into hCD1Tg/Rag^-/- or Rag^-/- mice 1 day before infection. Mice were euthanized at 4 week post-infection. (**A**) The number of bacteria in the lung, spleen and liver of individual mouse in each group. Horizontal bars represent the mean CFU counts ± SEM for each group. (**B, C**) Cells harvested from lungs of hCD1Tg/Rag^-/- or Rag^-/- mice were stimulated with un-pulsed or MA-pulsed DC and intracellular stained for the indicated cytokines (**B**) and CD107a expression (**C**). (**D**) Immunohistochemistry staining of anti-CD3 (brown cells) of the lung section from indicated groups. Pictures show granuloma area in infected lung tissues. (**E**) Bar graphs depict the mean and SEM of number of CD3^+ cells per mm^2 within granuloma areas (n=3–6 mice each group). ***p<0.001; **p<0.01; *p<0.05.

The following figure supplement is available for figure 7:

**Figure supplement 1.** Adoptive transfer of listeria LemA-specific M3-restricted D7 T cells does not confer protection against Mtb infection.

*2007*; *Cho et al., 2011*). Interestingly, MHC Ib H2-M3-restricted T cells could be selected by both HCs and TECs, however, only HC-selected T cells exhibited a pre-activated phenotype (*Cho et al., 2011*). Additionally, CD1d-restricted iNKT cells and CD1b-autoreactive T cells, which are also pre-

activated, are selected by HCs (*Bendelac, 1995*; *Coles and Raulet, 2000*; *Li et al., 2011*). Surprisingly, DN1 T cells, though phenotypically naïve, were most efficiently selected by HCs. This finding suggests that HC-mediated selection could lead to the development of two distinct phenotypes of group 1 CD1-restricted T cells. However, the mechanism behind this dichotomous selection process is unknown. Factors like TCR-CD1b avidity, the nature of the selecting lipid antigen(s) could all play a role in determining the phenotype of DN1 T cells. The fact that CD5 expression was significantly higher on DN1 T cells selected by HCs suggests that a stronger TCR-CD1b interaction ensued between HCs and DN1 T cells (*Azzam et al., 2001*). The strength of the interaction could be a reflection of the high CD1b expression on DP thymocytes compared to TECs. Although a significant proportion of DN1 T cells in DN1Tg/hCD1Tg mice are CD8$^+$, we did not detect differences in the activation kinetics and effector functions between CD8$^+$ or CD8$^-$ DN1 T cells. Also, HC-selected and TEC-selected DN1 T cells in bone marrow chimeric mice have similar proportions of CD8$^+$ T cells, suggesting that CD8 may not play a critical role in the selection and function of group 1 CD1-restricted T cells.

From a naïve phenotype, DN1 T cells transformed to an activated state upon encountering Mtb-derived MA. When CD1b-expressing DCs were infected with Mtb, DN1 T cells effectively lowered bacterial burden. This response was dependent on the presence of CD1b on DCs. Additionally, when Mtb-infected BMDMs (which do not express CD1b and are known to be the natural reservoirs of Mtb) were co-cultured with uninfected CD1b-expressing DCs, DN1 T cells retained their protective capacity. In this in vitro Mtb infection system, BMDMs were washed extensively after infection to remove extracellular Mtb before addition of the DCs and DN1 T cells. This suggests that CD1b-expressing DCs are most likely capable of cross-presenting Mtb-lipid antigen from infected macrophages to DN1 T cells. This is consistent with a previous study, which showed that upon death of mycobacteria infected-macrophages; apoptotic vesicles containing Mtb antigens are taken up by DCs, which present these antigens to T cells (*Schaible et al., 2003*). When DN1 T cells were adoptively transferred to Mtb-infected mice, they were first activated in the lung draining MLN instead of the lungs. Recent studies have shown that, after aerosol infection of mice with Mtb, myeloid DCs become infected in the lung, and represent the predominant cells that contain Mtb in the MLN (*Wolf et al., 2007*). Since peripheral CD1b expression is limited to a subset of myeloid DCs (*Felio et al., 2009*), initial DN1 T cell activation could be mediated by infected CD1b-expressing DC or by uninfected DCs that cross-present MA from infected DCs migrating from the lungs. However, the events and processes leading to the presentation of MA by DCs in the MLN remain to be clearly elucidated.

The protective capacity of DN1 T cells was also demonstrated in vivo upon transferring effector DN1 T cells to Mtb infected group 1 CD1 expressing Rag$^{-/-}$ mice. Bacterial burden reduction of approximately five (lung) to ten (spleen and liver) fold was observed in various organs. This demonstrated that DN1 T cells were capable of providing systemic immunity against Mtb. Numerous studies have shown that conventional CD4$^+$ and CD8$^+$ T cells are critical for resistance to Mtb infection, though it's widely accepted that CD4$^+$ T cells play a more dominant role (*Flynn and Chan, 2001*; *Woodworth and Behar, 2006*). Interestingly, studies using adoptive transfer of transgenic CD4$^+$ T cells reactive to Mtb antigens have yielded mixed results. While Ag85B specific T cells did not confer protection against Mtb infection (*Reba et al., 2014*), ESAT-6 reactive CD4$^+$ T cells reduced bacterial burden by a hundred fold, which was dependent on the infectious dose (*Gallegos et al., 2008*). On the other hand, immune conventional CD8$^+$ T cells lowered Mtb CFU by ten fold in the lungs of infected mice (*Woodworth et al., 2008*). Recently, the role of CD1d-restricted iNKT cells has been more appreciated in the context of in vivo Mtb infection. For example, mice that lack CD1d, the only CD1 isoform expressed in mice, did not show any differences in bacterial burden upon Mtb infection compared to wild type mice (*Behar et al., 1999*). However, upon adoptive transfer of iNKT cells, bacterial burden was reduced five fold in the lungs, suggesting iNKT cells may play a protective role during infection (*Sada-Ovalle et al., 2008*). Based on these studies, the protective capacity of DN1 T cells appears to be comparable to that of conventional CD8$^+$ T cells and CD1d-restricted iNKT cells, suggesting an important role of CD1b-restricted T cells in anti-mycobacterial immunity.

Traditionally, T effector responses against Mtb have been associated with production of IFN-$\gamma$ by conventional T cells (*Behar, 2013*). Aside from cytokine-mediated protection, CD8$^+$ T cells are also capable of contributing to anti-mycobacterial immunity through their cytolytic activities. Even though T cell derived IFN-$\gamma$ has been associated with macrophage activation-induced control of Mtb, several

studies have shown that IFN-γ alone does not absolutely correlate with protection (*Torrado et al., 2011*; *Gallegos et al., 2011*). However, IFN-γ, TNF-α and IL-2 polyfunctional cytokine producing T cells have been linked to protection against mycobacteria (*Darrah et al., 2007*; *Aagaard et al., 2009*; *Aagaard et al., 2009*). The anti-mycobacterial immunity conferred by DN1 T cells was probably mediated in part by the concomitant secretion of cytokines (IFN-γ, TNF-α and IL-2) and cytotoxicity. These data suggest that DN1 T cells employ a two-pronged mechanism for contributing to anti-mycobacterial immunity. A recent study suggests that CD1d-restricted iNKT cells mediate anti-mycobacterial immunity by producing GM-CSF (*Rothchild et al., 2014*). However, activated DN1 T cells did not seem to produce significant amounts of GM-CSF, suggesting that the protective effects of DN1 T cells may not be mediated by this cytokine.

Although DN1 T cells shared similar mechanisms of imparting protective immunity against Mtb with conventional T cells, DN1 T cells showed a faster activation and proliferation kinetics than Ag85B-specific CD4$^+$ T cells when they were co-transferred to Mtb-infected mice. Several mechanisms could contribute to the distinct kinetics of DN1 T cells and CD4$^+$ T cells during Mtb infection. Such mechanisms include: the selecting cell type in the thymus; availability of microbial antigen during infection; expression levels of antigen-presenting molecules and efficiency of antigen loading onto antigen presenting molecules. Interestingly, a previous study has shown that rapid DC maturation after Mtb infection induces MHC II trafficking to the plasma membrane without efficient antigen loading. On the other hand, CD1b, which continuously surveys the phagosome for antigens, is properly loaded with Mtb lipid antigen (*Hava et al., 2008*). This leads to earlier antigen presentation by CD1b compared to MHC II molecules. This phenomenon could at least in part explain why priming of DN1 T cells occurs earlier than Ag85B-specific MHC class II-restricted CD4$^+$ T cells during Mtb infection.

Even though Mtb lipid-specific group 1 CD1-restricted T cells have been implicated to play a protective role during Mtb infection, this is the first study to directly demonstrate their anti-mycobacterial activity in an in vivo infection setting. The secretion of various cytokines, the cytotoxic capacity, and the ability to be recruited to the site of infection likely contribute to the protective effect of MA-specific T cells during Mtb infection. Additionally, these T cells are activated and proliferate earlier than conventional CD4$^+$ T cells. Given these promising findings, it would be worthwhile to explore whether a multi-subunit vaccine, containing both Mtb protein and lipid antigens that activate both conventional and group 1 CD1-restricted T cells, is more effective against Mtb. Furthermore, like DN1 T cells, a substantial proportion of group 1 CD1-restricted T cells are CD4 negative (*Montamat-Sicotte et al., 2011*; *Gong et al., 1998*), and would not be affected by HIV infection. Thus, harnessing this subset of group 1 CD1-restricted T cells in vaccination strategies could prove to be particularly beneficial for HIV-infected patients.

## Materials and methods

### Ethics statement

This study was carried out in strict accordance with the recommendations in the Guide for the Care and Use of Laboratory Animals of the National Institutes of Health. The protocol was approved by the Animal Care and Use Committee of the Northwestern University (Protocol number: 2012–1736).

### Generation of DN1 TCR Tg mice and other mice used in this study

To generate DN1Tg mice, the variable regions of *TCR* genes were amplified from *DN1 TCR* plasmids (*Grant et al., 1999*) using the using the following primer pairs: *TRAV13-2*-for-5'-CCAAGATCTACCATGGCAGGCATTCG-3' and *TRAJ57*-rev-5'-CACAGCAGGTTCTGGGTTCTGGATATATG-3, and *TRBV5-1*-for-5'-CCTGGCCCAATGGGCTCCAG-3' and *TRBJ2̃ 7*-rev-5'-GGAGTCACATTTCTCAGGTCCTCTGTGAC. The constant regions of mouse *TCR* were amplified from a *HJ1 TCR* plasmid (*Li et al., 2011*), which encodes murine TCR α and b chain linked together by a 2A peptide, using the following primer pairs: *Cα*-for-5'-CATATATCCAGAACCCAGAACCTGCTGTG-3' and *2A*end-rev-5'-CTGGAGCCCATT GGGCCAGG-3', and *Cβ*-for-5'GTCACAGAGGACCTGAGAAATGTGACTCC-3' and *Cβ*-rev-5'-GCGTCGCTCGAGTCAGGAATTTTTTTTC-3'. Recombinant PCR was performed to connect human *VαJα*, murine *Cα-2A*, human *VβDβJβ* and murine *Cβ* fragments. Amplified *TCR* fragment was cloned into the *VA hCD2* cassette vector (*Zhumabekov et al., 1995*). DNA fragment

containing promoter and locus control regions of human *CD2* and chimeric *DN1* TCR was excised from the vector by *NotI /SalI* digestion and injected into fertilized B6 oocytes by the Northwestern Transgenic Core Facility. The presence of *DN1 TCR* in the genomic DNA of transgenic mice was examined by PCR using the *TRAV13-2*-for and *TRAJ57*-rev primers. DN1Tg mice were further crossed onto hCD1Tg (*Felio et al., 2009*) and Rag$^{-/-}$ backgrounds. P25Tg mice, expressing TCR specific for I-A$^b$/Mtb Ag85B peptide (aa 240–254), were purchased from Jackson Lab and further crossed onto Rag$^{-/-}$ background. D7Tg mice, expressing TCR specific for H2-M3/LemA peptide, were generated in our lab and were crossed onto the Rag$^{-/-}$ background (*Chiu et al., 1999*).

## Cell preparations and flow cytometry

Single-cell suspensions from organs were prepared and stained with the appropriate combinations of mAbs as described previously (*Felio et al., 2009*). All mAbs were purchased from BioLegend (San Diego, CA) and BD Biosciences (San Jose, CA). PerCPCy5.5-conjugated anti-TCRβ, PerCP-conjugated anti-CD45.2, APC-conjugated anti-hVβ5.1, Pacific blue-conjugated anti-CD4, IFN-γ and CD107a, FITC-conjugated anti-CD44, CD122 and TNF-α, PE-conjugated anti-CD69, PLZF, CD1b, CD5 and IL-2, PeCy7-conjugated anti-CD62L and CD44, and BV510 conjugated anti-CD8 mAbs were used. PLZF expression on thymocytes was analyzed via intracellular staining using the FoxP3 staining buffer set (eBioscience, San Diego, CA). For CD107a staining, anti-CD107a mAb was added during in vitro stimulation. For intracellular cytokine staining, cells were fixed with 4% paraformaldehyde, permeabilized with 0.1% saponin, and then stained with anti-cytokine mAbs. Flow cytometry was performed with a FACSCanto II and analyzed using FlowJo software. Fluorescence Minus One (FMO) was used to identify gating boundaries. Thymic stromal cells were prepared by digesting thymus with collagenase IV (1 mg/ml) and DNase (30 µg/ml). BMDCs and BMDMs were differentiated from mouse bone marrow progenitors as previously described (*Chun et al., 2003*). To generate DN1 effector cells, naïve DN1 T cells were cultured with irradiated hCD1Tg BMDCs pulsed with 10 µg/ml of mycolic acids (Sigma) for 1–2 weeks.

## Generation of BM chimeras

$1 \times 10^7$ BM cells from DN1Tg/Rag$^{-/-}$ and DN1Tg/hCD1Tg/Rag$^{-/-}$ mice were injected i.v. into recipients that were irradiated with 1000 rad one day before. Lymphocytes isolated from recipient mice were analyzed by flow cytometry 5–6 weeks after transfer.

## IFN-γ ELISPOT assay

DN1 T cells were purified from the spleen and lymph nodes from DN1Tg/hCD1Tg/Rag$^{-/-}$ mice through depletion of CD11c$^+$, DX5$^+$, CD11b$^+$ and MHC II$^+$ cells using mAb and streptavidin magnetic beads (Miltenyi Biotec). BMDCs from WT and hCD1Tg mice were pulsed with 10 µg/ml MA (Sigma, St Louis, MO overnight. ELISPOT assay was performed as previously described (*Felio et al., 2009*). IFN-γ producing cells were quantitated using an ImmunoSpot reader (Cellular Technology Ltd., Shaker Heights, OH).

## Cytotoxicity assay

DN1 effectors were established by stimulating naïve DN1 T cells isolated from pooled peripheral lymph nodes of DN1Tg/hCD1Tg/Rag$^{-/-}$ mice with MA-pulsed hCD1Tg BMDCs for 7 days in supplemented Mischell Dutton medium with IL-2 (20 U/mL). Target cells were labeled with 50 µCi [$^{51}$Cr] sodium chromate for 1 hr and cultured with effectors for 4 hr at 37°C. The percentage of specific lysis was calculated as (experimental release-spontaneous release)/(maximum release-spontaneous release) $\times 100$.

## Mtb in vitro culture and infection

Mtb H37Rv was grown and prepared as previously described (*Felio et al., 2009*). Mtb were added to BMDCs and BMDMs at an effective multiplicity of infection (MOI) of 1 for CFU experiments (or 5 for ELISA and FACS assays) for 2 hr. Cultures were washed three times and treated with 20 ng/ml gentamycin for 2 hr to remove extracellular bacteria. DN1 T cells purified from DN1Tg/hCD1Tg/Rag$^{-/-}$ mice ($3 \times 10^5$ cells/well) were added one day after infection. Culture supernatants were collected after 48 hr of co-culture and cytokines in the supernatant were detected using Cytometric Bead

Array or by ELISA. For CFU measurement, cells were lysed at day 7 post infection with 1% Triton X-100 in PBS, lysate were plated in serial dilutions on Middlebrook 7H11 agar plates and cultured at 37°C for 2–3 weeks. Neutralizing mAb for IFN-γ and TNF-α were added on day 1 post infection at 10 µg/ml.

### In vivo Mtb aerosol infection and adoptive cell transfers

For Mtb aerosol infection, mice were infected with 100–200 CFU using a nose-only aerosol exposure chamber (In-Tox Products, Moriarty, NM). At indicated time-points after infection, bacterial burdens in lungs and spleens were determined by plating serial dilutions of homogenate on Middlebrook 7H11 agar plates. For in vivo protection experiments, $5–7\times10^6$ DN1 or D7 effector cells were injected i.v. into recipients 1 day before infection. For other adoptive transfer experiments, $3–5\times10^6$ CFSE-labeled naïve DN1 T cells (from DN1Tg/hCD1Tg/Rag$^{-/-}$ mice) or CellTrace$^{TM}$ Violet-labeled P25 T cells (from P25Tg/Rag$^{-/-}$ mice) were transferred into recipients that were infected 7 days before.

### Immunohistochemistry and microscopy

Lungs from Mtb infected mice were fixed in 10% neutral buffered formalin and embedded in paraffin. 5 µm sections were acquired and deparaffinized. Sections were blocked with 5% normal donkey serum and incubated with primary antibody (CD3, Abcam-16669, UK). Subsequently, sections were washed and stained with secondary antibody followed by incubation with Vectastain (Vector Laboratories, Burlingame, CA). Lastly, sections were incubated with Biotinyl Tyramide working solution and streptavidin-HRP. The slides were scanned using the digital TissueFAXS imaging system equipped with a Zeiss Axio microscope (TissueGnostics GmbH, Austria). The area of granulomas and number of CD3 positive cells in the tissue sections were analyzed using HistoQuest image analysis software.

### Statistical Analysis

Statistical analysis was performed with the unpaired Student's $t$ test and one-way ANOVA followed by Bonferroni post-hoc test (for comparison of in vivo CFU data between different groups). All statistical analyses were performed with Prism software. Statistically significant differences are noted (***$p<0.001$; **$p<0.01$; *$p<0.05$).

## Acknowledgements

We thank the Northwestern University Transgenic Core facility for microinjection of *DN1 TCR* construct, Mouse Histology & Phenotyping Laboratory for histology services, Cell Imaging Facility for assisting with image analysis, Dr. Michael B Brenner (Harvard Medical School) for providing *DN1 TCR* plasmids, Dr. E Huesby (University of Massachusetts Medical School) for providing *hCD2* cassette vector, the NIH tetramer core facility for CD1d tetramer and Sharmila Shanmuganad for technical assistance. The authors have no competing financial interests.

## Additional information

### Funding

| Funder | Grant reference number | Author |
| --- | --- | --- |
| National Institutes of Health | AI057460 | Chyung-Ru Wang |
| National Institutes of Health | AI040310 | Chyung-Ru Wang |

The funders had no role in study design, data collection and interpretation, or the decision to submit the work for publication.

### Author contributions

JZ, Conception and design, Acquisition of data, Analysis and interpretation of data, Drafting or revising the article; SS, Acquisition of data, Analysis and interpretation of data, Drafting or revising the article; SS, YB, YH, Acquisition of data, Analysis and interpretation of data; SB, Analysis and

interpretation of data, Drafting or revising the article; CRW, Conception and design, Analysis and interpretation of data, Drafting or revising the article

## Ethics

Animal experimentation: This study was carried out in strict accordance with the recommendations in the Guide for the Care and Use of Laboratory Animals of the National Institutes of Health. The protocol was approved by the Animal Care and Use Committee of the Northwestern University (Protocol number: 2012-1736).

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
