## [Decision Letter]

Thank you for submitting your work entitled "Mycolic Acid-specific T Cells Protect Against *Mycobacterium tuberculosis* Infection In a Humanized Transgenic Mouse Model" for peer review at *eLife*. Your submission has been favorably evaluated by Tadatsugu Taniguchi (Senior editor) and three reviewers.

The following individuals responsible for the peer review of your submission have agreed to reveal their identity: Lucia Mori and Luc Van Kear (peer reviewers).

The reviewers have discussed the reviews with one another and the Senior editor has drafted this decision to help you prepare a revised submission.

Summary:

Thus far, in vivo evaluation of the biological function of class I CD1 molecules (i.e. CD1a, CD1b and CD1c) and CD1-restricted TCRs has been difficult because mice only express class II CD1 (i.e. CD1d). To address this problem, the authors generated mice that express a transgene encoding a CD1b-resticted human/mouse chimeric TCR that detects mycolic acid (DN1 Tg). Taking advantage of previously generated human CD1b Tg mice, the authors show that T cells expressing the DN1 transgene are positively selected in the thymus by hematopoietic cells and respond to DC pulsed with mycolic acid or infected with Mtb. While in vitro derived macrophages do not activate DN1 Tg T cells alone, they do in the presence of uninfected DC, suggesting that the mycolic acid is transferred from macrophages to DC, which can activate T cell responses. The new mouse line allowed to study the requirements for positive selection of human CD1b-restricted T cells and to explore the capacity of these transgenic T cells to protect mice from *M. tuberculosis* infection. The new data confirm the important finding previously obtained from the same group (using a mouse line Tg for a CD1b-autoreactive human TCR), that CD1b-expressing hematopoietic cells are the main drivers of thymic positive selection. Overall, these findings provide the first clear evidence that group 1 CD1-restricted T cells can protect against *M. tuberculosis* infection. It also provides an experimental model that will be helpful to test the efficacy of lipid-based vaccines. The work therefore represents an important advance that will move the field forward.

Essential revisions:

1) The authors should take advantage of this model to see whether vaccination with mycolic acid or with DC pulsed with mycolic acid provides resistance to subsequent challenge with aerosolic Mtb.

2) The authors showed that although to a lower extent, also TEC were able to support selection of human DN1Tg T cells and they also showed that small numbers of human DN1Tg T cells could be found in mice in the absence of human CD1 molecules. These findings suggest that perhaps mouse CD1d could be responsible for this selection. Adoptive transfer experiments into mice with CD1d^-/-^ background could clarify this issue.

3) The functional phenotype of DN1Tg T cells showed that these cells secreted high amounts of cytokines in response to hCD1Tg BMDCs infected with *M. tuberculosis*, but also significant amounts of TNFα, IL-6 and IL-17 in response to infected WT DCs. Could the author comment on that?

4) The important aspect of protection against *M. tuberculosis* infection was addressed using adoptive transfer of DN1 Tg T cells into hCD1Tg Rag^-/-^ mice subsequently infected. The experimental design misses two important controls in order to unequivocally demonstrate that protection is mediated by DN1 Tg cells: 1) adoptive transfer of irrelevant T cells into hCD1Tg Rag^-/-^ mice and 2) adoptive transfer of DN1 Tg T cells cells into WT mice. As a further control, as the authors elegantly used P25 Tg T cells (specific for Ag85B Mtb peptide) for comparison of classical vs. CD1-restricted T cells, I would suggest to use these cells as well. In addition, why not looking at protection directly infecting with *M. tuberculosis* the DN1Tg/hCD1Tg double Tg mice?

[Editors' note: further revisions were requested prior to acceptance, as described below.]

Thank you for resubmitting your work entitled "Mycolic Acid-specific T Cells Protect Against *Mycobacterium tuberculosis* Infection In a Humanized Transgenic Mouse Model" for further consideration at *eLife*. Your revised article has been favorably evaluated by Tadatsugu Taniguchi (Senior editor) and two reviewers. The manuscript has been greatly improved but we would like you to address the following point:

In the rebuttal letter you mention that DN1Tg/hCD1Tg double transgenic mice have an unusual T cell repertoire. We think that it is important to share this observation with the scientific community. Perhaps the finding reported in Figure 3 on the fact that large amounts of IL-17 are produced by T cells in response to Mtb-infected DC independently from hCD1b, could be explained by such T cell repertoire. We would like you to comment on these findings in the text.

---

## [Author Response]

*Essential revisions: 1) The authors should take advantage of this model to see whether vaccination with mycolic acid or with DC pulsed with mycolic acid provides resistance to subsequent challenge with aerosolic Mtb.*

In our preliminary studies, hCD1Tg mice vaccinated with Mtb lipid pulsed-DCs and subsequently challenged with aerosolic Mtb had lower bacterial load in the spleen but not in the lung. Given this preliminary data, my lab is now exploring the option of using nanoparticle loaded with mycolic acid as an alternative and more effective form of vaccination. The mycolic acid-loaded nanoparticles are being delivered to the mice intranasally so that antigens are present at the site of infection. However, it is extremely time consuming to optimize the vaccination strategies. Therefore, we would like to address the efficacy of lipid-based TB subunit vaccines more thoroughly in a follow-up study.

*2) The authors showed that although to a lower extent, also TEC were able to support selection of human DN1Tg T cells and they also showed that small numbers of human DN1Tg T cells could be found in mice in the absence of human CD1 molecules. These findings suggest that perhaps mouse CD1d could be responsible for this selection. Adoptive transfer experiments into mice with CD1d^-/-^ background could clarify this issue.*

We agree with the reviewers’ comments that CD1d could potentially act as a selecting molecule for DN1 T cells in the absence of CD1b. To address this issue, we compared the percentage of DN1 T cells in the spleen and thymus of DN1Tg/hCD1Tg (CD1d^+^), DN1Tg/hCD1Tg/CD1d^-/-^, DN1Tg(CD1d^+^) and DN1Tg/CD1d^-/-^ mice. We found that the percentage of DN1 T cells were comparable in DN1Tg/hCD1Tg and DN1Tg/hCD1Tg/CD1d^-/-^ mice. In addition, DN1 T cells were barely detectable in the thymus and spleen of DN1Tg and DN1Tg/CD1d^-/-^ mice. These data suggest that CD1d does not contribute to the thymic selection of DN1 T cells. We have included these data as Figure 2—figure supplement 1. A paragraph describing these results can be found in the subsection “CD1b-expressing hematopoietic cells (HCs) most efficiently select DN1 T cells”.

*3) The functional phenotype of DN1Tg T cells showed that these cells secreted high amounts of cytokines in response to hCD1Tg BMDCs infected with* M. tuberculosis*, but also significant amounts of TNFα, IL-6 and IL-17 in response to infected WT DCs. Could the author comment on that?*

Infected DCs (both hCD1Tg and WT) alone secrete minimal TNF-α, IL-6 and IL-17. Therefore, DN1 T cells in the presence of infected DCs secrete these cytokines. While TNF-α and IL-6 secretion by DN1 T cells is partially dependent on the presence of CD1b, IL-17 secretion is CD1b independent. We speculate that co-stimulatory molecules and/or cytokines induced by Mtb-infected DCs can stimulate DN1 T cells to secrete these cytokines independent of TCR-CD1b interaction.

*4) The important aspect of protection against* M. tuberculosis *infection was addressed using adoptive transfer of DN1 Tg T cells into hCD1Tg Rag^-/-^ mice subsequently infected. The experimental design misses two important controls in order to unequivocally demonstrate that protection is mediated by DN1 Tg cells: 1) adoptive transfer of irrelevant T cells into hCD1Tg Rag^-/-^ mice and 2) adoptive transfer of DN1 Tg T cells cells into WT mice. As a further control, as the authors elegantly used P25 Tg T cells (specific for Ag85B Mtb peptide) for comparison of classical vs. CD1-restricted T cells, I would suggest to use these cells as well. In addition, why not looking at protection directly infecting with* M. tuberculosis *the DN1Tg/hCD1Tg double Tg mice?*

According to the reviewers’ suggestion, we have included CFU data from adoptive transfer of DN1 T cells into Mtb-infected (WT, hCD1Tg^-^) Rag^-/-^ mice and demonstrated that the protective effect of DN1 T cells is hCD1Tg-dependent (revised Figure 7). We have also showed that substantial numbers of DN1 T cells were present in granuloma areas of infected hCD1Tg/Rag^-/-^ mice, but not Rag^-/-^ mice (revised Figure 7). As an additional control, we adoptively transferred D7 Tg T cells, specific for M3-Listeria LemA peptide, to hCD1Tg/Rag^-/-^ mice. Upon infection with Mtb, these T cells did not confer significant protection when compared to non-T cell transferred mice. We have included this data as Figure 7—figure supplement 1.

Due to the limited number of hCD1Tg/Rag^-/-^ mice currently available in the lab, we were not able to include P25 Tg T cells in our protection study. However, a study from Rebaet al*.* has shown that adoptive transfer of P25 Tg T cells did not confer protection against Mtb infection (Reference #37).

DN1Tg/hCD1Tg mice were not directly infected for protection assays because the T cell repertoire in these mice is substantially different from that of hCD1Tg mice (e.g. DN1Tg/hCD1Tg mice have decreased percentages of MHC I and MHC II-restricted T cells and NKT cells). This could introduce an unwanted variable, complicating data interpretation.

[Editors' note: further revisions were requested prior to acceptance, as described below.]

*In the rebuttal letter you mention that DN1Tg/hCD1Tg double transgenic mice have an unusual T cell repertoire. We think that it is important to share this observation with the scientific community. Perhaps the finding reported in Figure 3 on the fact that large amounts of IL-17 are produced by T cells in response to Mtb-infected DC independently from hCD1b, could be explained by such T cell repertoire. We would like you to comment on these findings in the text.*

We would like to clarify that experiments involving the characterization and assessment of DN1 T cells during Mtb infection have all been performed with DN1 T cells isolated from DN1Tg/hCD1Tg/Rag^-/-^ mice. This would eliminate the expression of endogenous TCR that may interfere with the analysis of DN1 T cells. Therefore, we do not anticipate the IL-17 present in the DN1 T cell and Mtb infected DC co-cultures to be a result of the unusual TCR repertoire of DN1Tg/hCD1Tg mice. We speculate that co-stimulatory molecules and/or cytokines induced by Mtb-infected DCs can stimulate DN1 T cells to secrete IL-17 independent of TCR-CD1b interaction (subsection “DN1 T cells exhibit effector functions in response to mycolic acid stimulation and Mtb infection”). This explanation has been included in the Results section. The use of DN1Tg/hCD1Tg mice in the Rag^-/-^ background has been mentioned at the beginning of the Results section of our manuscript. We have now included this information in the figure legends and Materials and methods section in this version of the manuscript. In the previous review, one of the reviewers had suggested that we perform the DN1 protection assay by directly infecting DN1Tg/hCD1Tg mice. By this comment, we assumed that the reviewer wanted us to infect DN1Tg/hCD1Tg mice in the Rag^+/+^ background and compare it to hCD1Tg mice in the Rag^+/+^ background, which have an intact immune system. Since these mice have endogenous TCR rearrangement aside from the expression of DN1Tg, we explained why the use of DN1Tg/hCD1Tg mice in the Rag^+/+^ background might complicate our conclusions. In addition, we found the CD4:CD8 T cell ratio in DN1Tg/hCD1Tg mice in the Rag^+/+^ background was altered compared to hCD1Tg mice in the Rag^+/+^ background. Also, the percentage of TCRβ^+^ NK1.1^+^ T cells was decreased in these mice. The above-mentioned observations are what we meant by an unusual TCR repertoire. We hope this explanation clarifies the questions raised by the reviewers.